# Advancing Liver Cancer Prevention for American Indian Populations in Arizona: An Integrative Review

**DOI:** 10.3390/ijerph19063268

**Published:** 2022-03-10

**Authors:** Timian M. Godfrey, Edgar A. Villavicencio, Kimberly Barra, Priscilla R. Sanderson, Kimberly Shea, Xiaoxiao Sun, David O. Garcia

**Affiliations:** 1College of Nursing, University of Arizona, Tucson, AZ 85721, USA; timiangodfrey@arizona.edu (T.M.G.); kshea@arizona.edu (K.S.); 2Mel and Enid Zuckerman College of Public Health, University of Arizona, Tucson, AZ 85724, USA; evillavicencio@arizona.edu (E.A.V.); xiaosun@arizona.edu (X.S.); 3A.T. Still University School of Osteopathic Medicine, Mesa, AZ 85206, USA; sa203705@atsu.edu; 4College of Health and Human Services, Northern Arizona University, Flagstaff, AZ 86011, USA; priscilla.sanderson@nau.edu

**Keywords:** Native Americans, health equity, healthcare disparities, liver diseases, liver cancer, preventative health services, disease prevention, social determinants of health

## Abstract

Liver cancer is a highly fatal condition disproportionately impacting American Indian populations. A thorough understanding of the existing literature is needed to inform region-specific liver cancer prevention efforts for American Indian people. This integrative review explores extant literature relevant to liver cancer in American Indian populations in Arizona and identifies factors of structural inequality affecting these groups. The Preferred Reporting Items for Systematic Reviews and Meta-Analyses guidelines informed the methodology, and a literature search was conducted in PubMed, EMBASE, CINAHL, and PsycInfo for articles including Arizona American Indian adults and liver disease outcomes. Seven articles met the inclusion criteria in the final review. Five of the studies used an observational study design with secondary analysis. One article used a quasiexperimental approach, and another employed a community-engagement method resulting in policy change. The results revealed a lack of empirical evidence on liver cancer prevention, treatment, and health interventions for American Indian populations in Arizona. Research is needed to evaluate the high rates of liver disease and cancer to inform culturally relevant interventions for liver cancer prevention. Community-engaged research that addresses structural inequality is a promising approach to improve inequities in liver cancer for American Indian people.

## 1. Introduction

Hepatocellular carcinoma (HCC) and intrahepatic bile duct cancer (cholangiocarcinoma), hereafter referred to as liver cancer, are highly lethal conditions with rapidly increasing mortality rates across the globe [1,2]. There is a 20% and 9% survival rate 5 years after receiving a HCC and intrahepatic bile duct cancer diagnosis, respectively [1,3]. Death rates from liver cancer outpace all other types of cancer [2]. These alarming statistics only worsen for American Indian populations. Liver cancer death rates for American Indian people (11.9 per 100,000) are more than double those of non-Hispanic Whites (5.5 per 100,000) [2]. The disparities in liver cancer can be linked to the disproportionate rates of major risk factors, such as obesity and type 2 diabetes (T2D), that are largely attributed to structural and systemic inequities [2,4]. To address liver cancer inequity for American Indian populations, there must first be an understanding of the existing data and scientific literature to inform cancer-control planning, early detection, and prevention efforts. However, the available literature and research studies investigating factors contributing to liver disease, and progression to cancer, are primarily conducted within Eurocentric populations [4]. Limited existing evidence contributes to the knowledge gap for liver cancer mitigation efforts in American Indian populations and bears the potential to increase cancer health disparities.

Further complicating matters, nationally aggregated data do not adequately portray the pertinent variation in significance between American Indian groups and the geographic differences in disparities, liver cancer incidence rates, and changes over time [5]. There are 574 federally recognized American Indian and Alaska Native Nations (also referred to as tribes, bands, pueblos, communities and native villages), each with a distinct culture, language, history, and social determinants of health shaping socioecological factors impacting liver disease. Health disparities research not only advocates for the comprehension of unique circumstances shaping existing inequities in marginalized populations but encourages further understanding into factors determining health outcomes for specific communities that may get lost in larger population health data without regard to local context [6,7]. To address health disparities in American Indian populations and advance health equity, evidence highlighting regional specific information can more accurately inform liver cancer prevention and control in American Indian communities. Arizona is richly concentrated and home to 22 federally recognized American Indian tribes. Arizona also has the third largest American Indian population count in the US with tribal lands comprising nearly 28% of the state’s land mass [8]. American Indian communities are important in the state and health researchers must be informed of local significance to develop high-impact interventions that appropriately address health disparities. However, the extent of liver disease in American Indian populations in Arizona is not well understood. For this reason, an interdisciplinary investigative team based in Arizona aimed to synthesize the body of available literature on liver disease and liver cancer amongst American Indian adults in Arizona to better inform future cancer prevention research endeavors.

This article will focus on Arizona-based American Indian populations to inform the larger scientific community by conducting a culturally tailored review. This targeted review may provide practices that could be adapted for other high-risk indigenous populations around the world. The objective of this article is to summarize extant literature relevant to liver disease, liver cancer, and American Indian populations in the Southwest region of the United States (US). Gaps in knowledge in liver cancer prevention will be identified, along with thorough discussion of structural inequities disproportionately affecting liver cancer for American Indian people.

Globally, liver cancer is the sixth most diagnosed type of cancer and the fourth leading cause of cancer-related mortality [9]. In 2018, the prevalence of global liver cancer burden was reflected in an estimated 841,000 cases and 782,000 deaths [10]. Risk factors for liver cancer include dietary exposure to aflatoxin, smoking, excessive alcohol intake, and nonalcoholic fatty liver disease (NAFLD) [11]. The prevalence of NAFLD among adults is estimated to be 25% worldwide [12]. Estimates in the prevalence of NAFLD varies between and within regions of the world, with the highest prevalence in the Middle East (32%) and South America (30%) and the lowest in Africa (13%) [13]. In the US, since 1980, liver cancer incidence rates have tripled and mortality rates have more than doubled [2,4]. In 2021, there were an estimated 42,230 new cases of liver cancer (29,890 in men and 12,340 in women) with an estimated 30,230 people (20,300 men and 9930 women) predicted to die from this condition [1]. Accompanying the high fatality rate is an approximate 19 years of life lost per death because a diagnosis commonly occurs at a young age (median = 63 years) [14]. This historical rise in liver cancer rates in the US is likely attributed to the simultaneous increase in rates of obesity and T2D [15,16,17]. Obesity is strongly linked to a variety of health conditions related to personal and social burden, including liver cancer [18,19,20]. Moreover, emerging evidence reports lifestyle related conditions such as NAFLD and metabolic syndrome, in which obesity and T2D are primary risk factors, are replacing cirrhosis, viral- and alcohol-related liver disease as the major forms of liver pathology leading to cancer [16,17]. Though American Indian populations experience some of the highest prevalence of obesity-related diseases in the US, when compared to all other racial and ethnic subpopulations [21,22,23], there are no longitudinal studies focused on early liver disease detection for American Indian, Alaska Native and Indigenous populations. This hinders national and global action on liver disease prevention and treatment options for these high-risk communities.

Although advances in liver cancer research have informed cancer prevention efforts and the incidence of liver cancer in the general US population is starting to level off, this promising trend is not reflected in American Indian populations [24]. Chronic liver disease and malignancy is a growing burden on the health of American Indian people across the US and remains a leading cause of death [5,25]. In 2018, American Indian adults were 1.6 times more likely to be diagnosed with chronic liver disease compared to non-Hispanic White adults [26]. From 2014–2018, American Indian and Alaska Native men were almost twice as likely to have liver cancer as compared to non-Hispanic White men; while American Indian and Alaska Native women were 2.2 times more likely than non-Hispanic White women to be diagnosed with liver cancer [24]. Further, American Indian and Alaska Native women are twice as likely to die from liver cancer, as compared to non-Hispanic White women [24]. In fact, the overall mortality rate for American Indian and Alaska Native people from liver disease is nearly four times higher than the non-Hispanic White population in the Southwest US [24,26,27]. More specifically, in 2019, the risk profile of American Indian adults for chronic liver disease and cirrhosis in Arizona was 462.6% worse than the average rate of all residents, with an accompanying rate of 82.7 deaths per 100,000 individuals [28]. As concerning as these statistics are, these figures appear to be underestimated as available data for most tribal nations on chronic liver disease and cancer remains unavailable. Progress in understanding the mechanisms involved in liver disease progressing to cancer is substantially needed to reduce the severe liver cancer disparities for American Indian populations.

The objective of this integrative literature review is to address the current gap in knowledge on the specific components and outcomes of studies on liver disease and/or liver cancer prevention for American Indian communities in Arizona. The aim of this review is to identify current best evidence and specifically answer the following questions: (a) What are the incidence and prevalence of liver disease and/or liver cancer for American Indian populations in Arizona?; and (b) What treatment and prevention strategies have been implemented to address liver disease and/or liver cancer for American Indian adults in Arizona [29]?

## 2. Methods

### 2.1. Search Strategy

The Preferred Reporting Items for Systematic Reviews and Meta-Analyses (PRISMA) guidelines informed the review process. A literature search was conducted across four major academic databases, PubMed, EMBASE, Cumulative Index to Nursing and Allied Health Literature (CINAHL), and PsycInfo. Due to the research questions, there were no limitations on publication dates to endorse evaluation of extant literature. The search strategies in PubMed and CINAHL used the National Library of Medicine Medical subject headings (MeSH). The development of key terms and search strategies were done in collaboration with a librarian from the University of Arizona Health Sciences Library. All suggested variants of MeSH terms were included in database searches. A combination of the following key terms with Boolean phrases “AND”, “OR” were utilized in the search strategy: “Liver disease”, “Liver cancer”, “Liver neoplasms”, “Hepatocellular carcinoma”, “Nonalcoholic fatty liver disease”, “Liver dysfunction”, “American Indian”, “Native American”, “Indigenous persons”, “American Natives”. Detailed key terms for each database are provided in Table 1.

It is important to note that while the term American Indian was used for this manuscript due to its widespread use in the research literature, we recognize that there are more inclusive terms that can be used to describe our focused population. As mentioned, we included other terms in our searches such as Native American, Indigenous persons/people, and American Natives.

### 2.2. Screening Process

Article selection was performed by recording the numerical results obtained from each individual search, including individual keywords, and combined controlled vocabulary for each database in an Excel spreadsheet. This information was then organized in a table. Inclusion criteria included full-text and English language articles from peer-reviewed journals. Additional criteria for article selection included studies focused on Arizona American Indian populations, ages 18 years and older, and liver disease categories (NAFLD, HCC, cirrhosis, hepatitis, and alcoholic liver disease) as primary or secondary outcomes. Studies that focused on children and/or adolescents, studies that included participants that did not identify American Indian participants, and studies that did not have liver disease as a major outcome were excluded.

Search result files from each database were imported by K.B. into the Rayyan Systems Inc. platform, a collaboration and research tool to collect and organize search results, detect duplicates, and remove unrelated articles in a systematic matter. After removing duplicates in Rayyan, coauthor E.A.V. uploaded the remaining results into EndNote, a citation manager software, to detect any additional duplicates. After additional duplicates were removed in EndNote, the reference list was then uploaded back onto Rayyan where coauthors E.A.V. and K.B. independently screened the articles generated from the literature based on inclusion/exclusion criteria. To further categorize liver disease outcomes, the investigative team utilized a labeling system during the screening process which included the categories of: Hepatitis, HCC, NAFLD, Cirrhosis, and Alcohol-related Liver Disease (ALD). The team also utilized labels to categorize different geographic regions of the studies to identify the focus population, including North American Indigenous (Canada, Alaska, Mexico), US American Indian, Southwest US American Indian, and Arizona American Indian. Figure 1 depicts the study selection process.

### 2.3. Study Selection

The database search yielded 2167 results. Once duplicates were removed, a total of 1691 were screened for inclusion. To provide more context as to the type of results that were obtained from the searches, the following are the results from each of the different subcategories that were utilized when screening articles: Hepatitis (*n* = 87), North American Indigenous (Canada, Alaska, Mexico) (*n* = 76), US American Indian (*n* = 36), HCC (*n* = 28), Southwest US American Indian (*n* = 11), Arizona American Indian (*n* = 11), NAFLD (*n* = 9), Cirrhosis (*n* = 9), and ALD (*n* = 3). It is important to note that many articles had more than one subcategory assigned to them. A total of 1681 articles did not meet the inclusion criteria during the title and abstract review. From the remainder, 10 were included to be part of the full-text screening, but 3 were published abstracts and were, therefore, excluded. A total of 7 articles were selected for inclusion in the integrative review. Analysis was complicated by variation in study design and methodology. 

## 3. Results

Full-text review of included articles resulted in one article focused on alcohol-related liver cirrhosis [30], one on increased liver fat (hepatic steatosis) [31], two on hepatitis (A and C) [32,33], one on overall chronic liver disease [34], and two described liver mortality [35,36]. Publication years ranged from 1971 to 2018 and study populations varied from three articles based in Arizona tribal populations only [35], three based on tribal multiregional areas in which Arizona was one of the focus areas [34,36], and two based in surveillance systems primarily focused in American Indian/Alaska Native individuals as part of the analysis [30]. Communities included the Navajo, Hopi, Tohono O’odham, Salt River Pima Maricopa, Ak-Chin, and Gila River tribes, as well as an Indian Health Service medical center in Phoenix. Study characteristics are shared in Table 2.

Most of the included articles (*n* = 5) had an observational study design with secondary analyses conducted for epidemiological surveillance measures. Medical records were the primary source of data collection for 71% of the studies. One study used a quasiexperimental, pre–post-test design using biomedical specimens (i.e., tissue biopsy, blood samples) and radiographic imaging as measures [31]. Although medical records were also used to measure and inform a surveillance program, the study by Gachupin et al. (2018) involved an observational design that informed community engagement efforts resulting in policy change [32]. This was also the only study to reference a theoretical framework.

Multiple studies noted the issue of misclassified data in either the American Indian identity of the sample and/or diagnosis codes [34,35,36]. Due to this, it is likely that prevalence data for liver disease is underestimated in these studies. Also important is the distinction in findings between different American Indian communities. Kunitz et al. (1971) detected significant differences in distributions of liver cirrhosis between Hopi and Navajo communities in Arizona [30]. Lee et al. (1998) noted significant regional differences in liver disease and cirrhosis mortality rates between tribal communities in Arizona, Oklahoma, South Dakota, and North Dakota [36]. Disproportionate rates of varying liver disease states for American Indian adults compared to the general US population were reflected in four of the studies [30,34,35,36]. Additionally, cardiometabolic conditions were concomitant findings in three studies [31,34,36], with one study also including malignancy [35].

### Study Findings

In 1971, Kunitz and colleagues [30] attempted to investigate the presence of alcoholic liver cirrhosis patterns in Arizona’s Hopi Tribe and Navajo Nation using medical records in the U.S. Public Health Service hospitals and clinics. The authors concluded that cirrhosis was present in about 60% of men in both tribes for their sample, and overall, the Hopi community showed over four times higher liver cirrhosis death rates compared to the general US population, while the Navajo community showed a slightly lower liver cirrhosis death rate when adjusting for age.

Two subsequent studies reported on community mortality surveillance in the Gila River Indian Community (GRIC) from 1975 to 1988. In 1990, Sievers et al. [35] presented mortality rates and causes of death based on available medical records and death certificates obtained from the Arizona Department of Vital Statistics from 1975 to 1984. During this decade, 681 deaths of adults from the GRIC were reported, but death certificate data was only obtained for 677 cases. The article highlights chronic liver disease and cirrhosis as replacing cardiovascular diseases as the second leading cause of death for American Indian people residing in the GRIC. At the same time, the mortality rate for chronic liver disease and cirrhosis in the GRIC surpassed that of the US, with 7.8% vs. 1.5%.

A few years later, Lee et al. (1998) [36] studied all-cause mortality and cardiovascular disease mortality from three different U.S. Indian tribes. Arizona’s Pima, Maricopa, and Tohono O’odham people were represented in this cohort of the Strong Heart Study residing in the Gila River, Salt River, and Ak-Chin communities between 1984–1988. Results of this analysis showed that liver disease and cirrhosis in the sample represented 70% of deaths from digestive diseases in women and 57% in men. In terms of cancer, 41% of cancer deaths in Arizona were classified as “cancer of the digestive organs” in which the liver was included. Specific numbers in liver cancer were not reported in the article. Liver disease and cirrhosis mortality rate for men and women in Arizona was the highest among participants aged 45–54 and in men of all ages when compared to data from tribal communities in Oklahoma, South Dakota, and North Dakota.

It was not until 10 years later when two more articles were able to elaborate on liver conditions among American Indian communities residing in Arizona. Koska et al. (2008) [31] evaluated the association between large abdominal fat cells and elevated fat build up in visceral, hepatic, and intramyocellular tissue in American Indian adults who are obese (18–45 years) from Pima tribal communities and with ordinary glucose tolerance. Outcomes of this study highlighted a positive correlation found in fat cell size with body fat and increased accumulation of intrahepatic lipid content. Additionally, plasma adiponectin, diameter of fat cell, and visceral adipose tissue seemed to independently predict intrahepatic lipid levels according to a multivariate analysis. Koska’s work was also able to conclude that elevated fat in the liver may self-sufficiently be associated with low levels of fasting plasma adiponectin, obesity class, and elevated visceral adipose tissue levels with peripheral and hepatic insulin resistance.

During the same year, Bialek and colleagues (2008) [34] cross-sectional study addressing chronic liver disease prevalence in two American Indian medical centers serving in the Southwest of the US. One of these centers was at Phoenix Indian Medical Center which is managed by the Indian Health Service and provides comprehensive data on American Indian people residing in the area. Data was collected from 30,698 American Indian patients during 2000–2002, of which 4.9% (*n* = 1496) had chronic liver disease. Of those, 17.9% were considered in the stage of decompensated cirrhosis and other causes included 6.9% of patients with hepatitis C, 41.5% for alcohol, and 12.8% for NAFLD. Men had a higher prevalence of chronic liver disease, ALD, and chronic hepatitis C; while women had a higher prevalence for NAFLD. Male patients aged 40–49 had the highest prevalence of chronic liver disease, chronic hepatitis C, and alcohol-related chronic liver disease and females aged 50–59 had the highest prevalence for chronic liver disease and NAFLD. It is worth mentioning that cause of chronic liver disease was not determined for 26.1% (*n* = 391) of patients and many met at least one of the criteria for NAFLD, including 34.8% with diabetes and 59.6% with obesity. Due to this cause, it is likely that prevalence data for NAFLD is significantly underestimated.

Two articles were published a few years later emphasizing the epidemiology of hepatitis A and C. The first was in 2012 where Erhart and Ernst [33] investigated epidemiological trends of hepatitis A in Arizona between 1988–2007 given policies in vaccine implementation, promotion, and coverage. Hepatitis A cases were obtained from the Arizona Department of Health Services, local public health departments, and Indian Health Service facilities. The incidence of hepatitis A in Arizona dropped significantly from 58 cases per 100,000 in 1988 to 2 cases per 100,000 in 2007. American Indian people had the highest incidence of hepatitis A prior to 1996, with rates ranging from 67 to 444 cases per 100,000, until 2004 when they registered lower than other racial/ethnic groups. Following this work, Gachupin et al. (2018) [32] explored methodologies to implement and expand tribal-based hepatitis C initiatives to promote treatment and prevention strategies in a Southwest tribal Health Service Division. Through comprehensive and culturally competent outreach efforts, 85 of the 251 hepatitis C positive tribal members engaged in education, field-testing, referral, and follow-up interactions. This represented in an increase of more than 300% in patient retention, engagement, and monitoring confirming the uprising commitment of tribal communities to eradicate this communicable disease. 

## 4. Discussion

American Indian communities experience high rates of liver cancer incidence and mortality. Despite the clear need to develop effective liver cancer prevention strategies, this review has confirmed that a gap remains among American Indian populations and the scientific literature. The gap includes a lack of early screening and interventions addressing the burden of liver disease progressing to liver cancer for tribal communities in Arizona. Most of the included articles focused on outcomes related to alcohol-related cirrhosis, hepatitis, chronic liver disease, and liver mortality as part of surveillance systems. Although there was mention of the need to develop interventions addressing sociocultural factors contributing to disparate findings [35], there is a lack of discussion about the influence of social determinants of health and systemic inequities that affect liver cancer prevention and treatment efforts.

To better understand the larger context of liver cancer in American Indian populations, the influence of structural inequity on social determinants of health must be included in the scientific literature. Longstanding racism and discrimination have created structural inequities shaping social determinants of health that heavily influence contemporary lifestyle-related health disparities in liver cancer and result in poorer health outcomes for American Indian people [37,38,39,40,41]. This is particularly important because an estimated 71% of liver cancer cases are preventable, as most risk factors that are prevalent in American Indian communities (e.g., chronic viral hepatitis, cirrhosis, obesity, NAFLD, T2D, heavy alcohol use, tobacco use, etc.) are modifiable with lifestyle and behavioral interventions [42]. However, consequential effects of colonization have created barriers for American Indian communities to achieve health equity. The loss of ancestral lands, traditional agricultural practices, and water rights resultant from forced relocation and assimilation practices caused widespread reliance on nutritionally deficient commodity food programs which, in turn, created disproportionate rates of certain health conditions that confer higher risk for liver cancer (e.g., T2D, metabolic syndrome, and NAFLD) [15,16,17,37]. Still present problems such as food insecurity affect over 25% of American Indian and Alaska Native members on over 60 tribal reservations [43]. Also problematic is the lack of affordable and accessible healthy food sources on tribal lands that are essential in preventing the development of risk factors for liver cancer [44].

Further contributing to the burden of liver cancer in American Indian groups is severely underfunded systems that affect access to health services [45]. For example, each person in the Indian Health Service system (including tribal 638 healthcare facilities and urban healthcare facilities) receives less than half of what is allocated to the general US population (USD 4078 vs. USD 9726) [46,47]. Needless to say, chronic inadequate funding of the primary health system for American Indian and Alaska Native people directly impacts disparities in liver cancer and associated risk factors [45].

The issue of liver cancer for American Indian people may be more complex than what is conveyed in the literature. To precisely assess the state of liver cancer, updated data on the burden of liver disease for American Indian communities in Arizona is needed. For most of the studies in the review, data was greater than 10 years old. Additionally, misclassification of data likely caused underreporting of liver disease and cancer rates [48]. Similar to our findings, cause-specific mortality data is known to be limited due to errors in diagnosis and reporting of cause of death in other national studies [49,50,51]. This targeted review provides a model that can be adapted to expand the breadth of reviews in this area, particularly for other Indigenous populations.

Importantly, early detection of liver disease remains critical for health promotion and prevention of advanced stages of disease. As such, it is eminent to recognize the need of available and cost-effective treatment and screening alternatives for the prevention of advanced liver disease. Recent evidence suggests the use of nonenhanced magnetic resonance imaging (MRI) as a reliable method for HCC surveillance among high-risk individuals. MRI’s advantages include its advanced performance, short scan times, and the lack of contrast agent-associated risks [52]. This screening strategy is more robust compared to tomography or ultrasound-based modalities for the detection of HCC. MRI and other advanced imaging methods remain the gold standard for liver steatosis detection. However, noninvasive examinations may support the accuracy and advancements of chronic liver disease in hard-to-reach populations [53,54]. For instance, transient elastography (Fibroscan^®^) is a noninvasive tool to assess liver steatosis and fibrosis that can be performed in the outpatient clinics and communities with immediate results and excellent reproducibility [55]. In fact, our research team was among the first to use noninvasive transient elastography (FibroScan) to estimate the prevalence of NAFLD in a community-based sample of 307 Mexican-origin adults in Southern Arizona in which the overall estimated prevalence of NAFLD was 50% [56]. This effort focused on the early detection of liver disease, given that there are no pharmaceuticals available for NAFLD treatment and weight loss is the cornerstone treatment [57]. Similar community-based noninvasive early detection and screening strategies may prove to be beneficial for hard-to-reach Indigenous populations.

Although structural inequality is interconnected with liver cancer health disparities, the importance of the strength and resilience within American Indian communities must be emphasized as a social determinant of health [32]. When research efforts culturally tailor approaches to the needs of a community and practice community engagement and/or community-based participatory research (CBPR) principles, outcomes are improved [32,58,59]. While most American Indian adults may be aware of liver disease (particularly cirrhosis and hepatitis), far less may be known about the potential benefits of early screening and detection to reduce progression to more advanced liver disease. Early adoption of lifestyle behaviors addressing modifiable risk factors for liver cancer is a prevention strategy that does not require substantive resources. A large proportion of liver cancer deaths could be averted, and existing disparities could be dramatically reduced, through the targeted application of existing knowledge in prevention, early detection, and treatment. Research teams should develop community-centered approaches to accelerate liver cancer prevention for this high-risk group. The principles of CBPR research that addresses structural inequalities are a promising approach to improve the inequities in liver cancer for American Indian people. Future research should apply integrated scientific approaches, including CBPR methods such as developing community advisory committees (CACs) comprised of liver cancer survivors, caregivers, and those directly affected by liver disease prior to starting research related activities. The goal is to work collaboratively with community members and organizational representatives to refine research approaches, share ownership and decision making, and receive feedback on the development processes of interventions to advance policy or social changes. In addition, fostering professional education and development for American Indian students and faculty members throughout the research process is essential. This will serve as a catalyst for training American Indian students and faculty in interdisciplinary and translational research, a “gap” area where greater representation is essential to building the evidence toward reduced health disparities for American Indian subpopulations. Comprehensive approaches can create synergy and long-lasting sustainable partnerships between tribal communities and researchers to advance health equity in liver disease and liver cancer. It also will ensure steps are taken to develop culturally and clinically relevant intervention strategies to reduce liver cancer risk and accelerate research around liver cancer prevention and treatment for tribal communities.

Coalescing objectives and priorities with existing clinical and policy systems is another mechanism to share resources and attain mutual benefit. This has been demonstrated by efforts to reduce rates of liver cancer in the Cherokee Nation Health Services, who implemented an HCV testing policy that interfaced with clinical healthcare providers [50]. Screening for HCV significantly increased and 90% of the individuals treated were cured after HCV diagnosis. Another exemplar is the collaboration between Indian Health Service (including tribal 638 healthcare facilities and urban healthcare facilities) and Project ECHO (Extension for Community Healthcare Outcomes) [50]. Project ECHO has increased access to care by creating partnerships between specialists and healthcare providers in rural and under-served communities. Using similar approaches, future research teams could develop collaborations between public health, clinical, and policy groups to advance liver cancer control efforts and access to care in Arizona tribal communities. The COVID-19 pandemic has presented the opportunity to use telehealth services in rural and remote locations [60,61]. However, careful action must be taken because the quality of virtual care, when received, is not always patient-centered or culturally competent [62,63]. Partnering with American Indian communities is recommended to promote culturally competent policies and delivery of care in telehealth [62]. Therefore, more research is needed on the development, implementation and evaluation of culturally and clinically relevant telehealth programs in rural settings for American Indian populations, particularly for cancer screening and health literacy.

Though the integrated programs described can improve access to care and liver cancer preventative services, broader intervention and prevention strategies are needed to address disparities in liver cancer incidence and risk factors for American Indian communities in Arizona. A comprehensive approach would address the many factors that contribute to health disparities, including structural inequity. Culturally congruent, community-based interventions are necessary to support healthy behaviors and promote best outcomes. Future research may involve interdisciplinary teams and community partners to explore ways to improve liver cancer prevention strategies that promote health equity in liver cancer for American Indian communities by applying integrated scientific approaches and community-engaged research methods.

### Limitations

This review has several potential limitations. Limitations in the articles analyzed included small sample sizes, and lack of grounding in theory that was consistent across the studies reviewed. Generalizability of the findings is limited due to the scope of the review and intention to refine the population of focus. Studies were limited to specific geographic areas, types of data sources, and methods. Most studies relied on secondary analysis of existing data and surveillance. An additional limitation was the outdated sources included in this review. There is a substantial need for updated evidence to truly understand the incidence and prevalence of liver cancer in American Indian communities in Arizona. Finally, an assessment of methodological quality was not incorporated into the review process due to the heterogeneity of included studies.

## 5. Conclusions

American Indian populations are largely underrepresented in the liver cancer prevention literature, particularly when considering modifiable liver cancer risk factors. To date, there is scant scientific knowledge focused on behavioral, social, and cultural factors to promote engagement more effectively in early screening and detection for liver cancer in this high-risk adult population. Additionally, there is a lack of empirical evidence on both liver cancer prevention and treatment for American Indian communities in Arizona. Our findings suggest there is no prior research ensuring American Indian adults understand what liver cancer is, why they are at risk for developing it, and what they can do to stop the progression of liver disease or reverse their condition. Another challenge not addressed by the literature is the prevalence and effect of NAFLD and other obesity-related liver cancer risk factors, independent of alcohol consumption, for American Indian populations. Lastly, if risk factors are identified, there is no clear evidence how to provide treatment options to participants that may be accessible, particularly for those in rural remote settings, and culturally and clinically relevant. More research is needed to evaluate the high rates of liver disease and cancer to inform culturally relevant interventions for liver cancer prevention and treatment.

## Figures and Tables

**Figure 1 ijerph-19-03268-f001:**
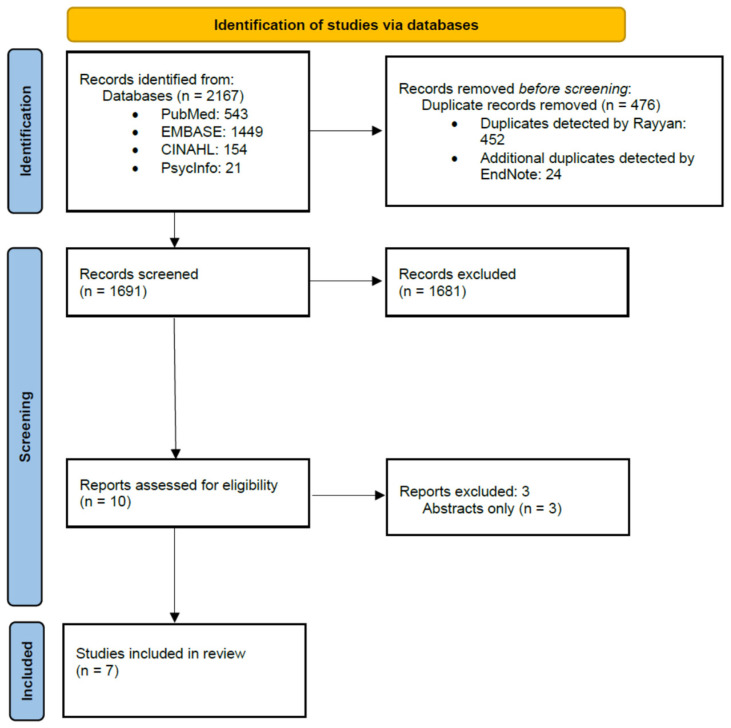
PRISMA Flow Diagram.

**Table 1 ijerph-19-03268-t001:** Search Strategy.

	PubMed	CINAHL	Embase	PsycInfo
Keywords	(“american indians” OR “american indian” OR “native americans” OR “native american” OR “indigenous persons” OR “indigenous peoples” OR “indigenous populations” OR “american natives” OR “american native” OR “American Natives” (MeSH Terms) OR “Indigenous Peoples” (MeSH Terms)) AND (“liver disease” OR “liver cancer” OR “liver neoplasms” OR “hepatocellular carcinoma” OR “nonalcoholic fatty liver disease” OR “non-alcoholic fatty liver disease” OR NAFLDOR “liver dysfunction” OR “Liver Diseases” (MeSH Terms) OR “Liver Neoplasms” (MeSH Terms))	(“american indians” OR “native americans” OR “indigenous peoples” OR “american natives” OR (MH “native americans+”) OR (MH “Medicine, Native American Traditional”)) AND (“liver disease” OR “liver cancer” OR “hepatocellular carcinoma” OR “nonalcoholic fatty liver disease” OR “non-alcoholic fatty liver disease” OR NAFLDOR“liver dysfunction” OR (MH “Liver Diseases+”) OR (MH “Liver Failure+”) OR (MH “Nonalcoholic Fatty Liver Disease”))	(‘indigenous people’/exp OR ‘native born’ OR ‘native people’ OR nativesOR‘american indian’/exp OR ‘american native’ OR ‘american natives’ OR amerindian OR ‘native american’ OR ‘north american indian’ OR ‘north american indians’) AND (‘liver disease’/exp OR ‘liver disease’ OR ‘liver diseases’ OR ‘hepatic disease’ OR ‘hepatic disorder’ OR ‘liver disorder’ OR ‘liver illness’ OR ‘chronic liver disease’ OR ‘fatty liver’/exp OR ‘fatty liver’ OR ‘liver fibrosis’ OR ‘nonalcoholic fatty liver’/exp OR ‘nonalcoholic fatty liver disease’ OR ‘liver cancer’/exp OR ‘liver cancer’ OR ‘liver cell carcinoma’/exp OR ‘liver cell carcinoma’)	(“american indians” OR “native americans” OR “indigenous peoples” OR “american natives” OR DE “American Indians” OR DE “Indigenous Populations” OR DE “Alaska Natives” OR DE “American Indians” OR DE “Inuit” OR DE “Pacific Islanders” OR DE “Alaska Natives” OR DE “Tribes”) AND (“liver disease” OR “liver cancer” OR hepatocellular carcinoma OR nonalcoholic fatty liver disease OR NAFLDOR“liver dysfunction” OR “DE “Liver Disorders” OR DE “Cirrhosis (Liver)” OR DE “Hepatitis” OR DE “Jaundice” OR DE “Cirrhosis (Liver)”OR DE “Neoplasms”))
Results	543 articles	154 articles	1449 articles	21 articles

**Table 2 ijerph-19-03268-t002:** Study characteristics of included articles.

Article	Author and Date	Evidence Type	Sample, Sample Size, and Setting	Study Findings That Help Answer the Research Question	Limitations	Outcomes/Other Findings
1	Kunitz et al. (1971)	Community surveillance retrieved from US Public Health Service hospitals and clinics	Epidemiology of alcoholic cirrhosis in Hopi Tribe (*n* = 25) and Navajo Nation (*n* = 91).	Cirrhosis was present in about 60% of men in both tribes. The Hopi community showed over four times higher liver cirrhosis death rates compared to the general US population. The Navajo community showed a slightly lower liver cirrhosis death rate when adjusting for age.	Sample sizes were relatively small. Comparisons between on and off each reservation was avoided.	Death percentages of liver cirrhosis among the Hopi tribe are over four times higher than the general US population. Among the Navajo, age-adjusted rate is only slightly less than in the general US population.
2	Sievers et al. (1990)	Retrospective, longitudinal study of diabetes and other disorders in the Gila River Indian Community (GRIC)	Death records collected from the National Institutes of Health comprising GRIC deaths that occurred in 1975–1984; death certificates were obtained for 677 of the 681 deaths.	Death percentages in the GRIC for liver disease and cirrhosis was one of the diseases that greatly exceeded that in the US. Mortality is much higher for the Pima Indians of the Gila River Indian Community than for the US all races.	The records reviewed were derived solely from information recorded on death certificates and did not include all available pertinent records to determine the most probable underlying cause of death.	Death-rate ratios are higher in all age categories for the GRIC Pima than for reported total American Indian population served by the Indian Health Service.
3	Lee et al. (1998)	Community mortality surveillance (from 1984–1988)	Three American Indian populations, aged 45–74 years, in Arizona, Oklahoma, and South/North Dakota. Arizona data: Men (*n* = 847), Women (*n* = 1242).	Liver disease and cirrhosis accounted for 57% of the deaths due to digestive disease in men and 70% in women. Men at the Arizona center had a much higher death rate due to liver disease and cirrhosis than at the other two centers.	Study was primarily focused on cardiovascular disease, although all-cause mortality was also examined.	Mortality rates during 1984–1988 among the three American Indian populations of the Strong Heart Study exceeded the general rates found in their respective states and in the US population.
4	Bialek (2008)	Cross-sectional prevalence study.	Study was conducted at medical centers serving predominantly American Indian populations in Arizona and California: Phoenix Indian Medical Center (PIMC) and Riverside San Bernardino County Indian Health Incorporated (RSBCIHI). Arizona center (PIMC): *n* = 30,698. California center (RSBCIHI): *n* = 6074. Overall study (both centers): N = 36,772.	CLD ^1^ is prevalent among American Indian patients in clinical care, with HCV ^2^ and ALD ^3^ being the two most common etiologies. Morbidity and mortality from CLD are likely to increase as the large number of patients infected with HCV during the 1980s develop clinical manifestations of cirrhosis.	Limited access to liver biopsy or radiologic evidence of steatohepatitis to confirm diagnosis of NAFLD ^4^. Diagnosis based on presence of obesity and diabetes, which was specific but not sensitive; therefore, the prevalence of NAFLD was likely underestimated. Study did not include all American Indians living in the study areas and only two sites were studied.	Of the 30,698 American Indian and Alaska Native adults who received care at PIMC during 10/2000-9/2002, 1496 (4.9%) had CLD; approximately 13% of patients with CLD had NAFLD as the primary cause. Prevalence of CLD, ALD, and HCV was higher among males than females, whereas NAFLD was more prevalent among females than males. Many of those with no etiology met at least some of the criteria for NAFLD, including 34.8% with diabetes and 59.6% who were obese. Eleven patients with CLD had HCC ^5^
5	Koska et al. (2008)	Quasiexperimental study: pretest, post-test design.	N = 53 Pima Indian individuals between ages of 18 and 45	Increased size of abdominal adipocytes predicts an increased liver fat content in obese individuals with normal glucose tolerance.	Average cell size might have been underestimated in subjects with the largest fat cells due to their increased propensity for disruption or lysis when treated by collagenase. Hepatic insulin sensitivity reflects primarily fasting endogenous glucose output in subjects with complete suppression of EGP (43% of the group) warranting some caution when interpreting the results on hepatic insulin sensitivity. Standard magnetic resonance imaging has a low sensitivity to detect small amounts of fat and low specificity to distinguish steatosis from other types of liver pathology.	In a multivariate analysis, plasma adiponectin, adipocyte diameter, and visceral adipose tissue (VAT) independently predicted intrahepatic lipid content (IHL). Low insulin-mediated glucose disposal was associated with low plasma adiponectin (*p* = 0.02) and high IHL (*p* = 0.0003), subcutaneous adipose tissue (SAT) (*p* = 0.02), and VAT (*p* = 0.04). High IHL was the only predictor of reduced insulin-mediated suppression of hepatic glucose production (*p* = 0.02) and the only independent predictor of insulin-mediated glucose disposal in a multivariate analysis.
6	Erhart and Ersnt (2012)	Retrospective surveillance reporting	N = 22,760; *n* = 1444 for American Indian individuals in the state of Arizona (1988–2007)	Incidence of hepatitis A in Arizona dropped significantly from 58 cases per 100,000 in 1988 to 2 cases per 100,000 in 2007.	There is potential for underreporting from federal Indian Health Service facilities in this study compared healthcare providers due to lack of mandated reporting.	Racial/ethnic disparities between American Indian and non-Hispanic White populations appear to be eliminated according to data.
7	Gachupin et al. (2018)	Surveillance, community engagement	N = 251 tribal members identified with HCV between 2009–2014	Expansion of a publicly supported disease identification and treatment services for HCV-positive clients. This enabled an integrated, structured and reliable system for the southwest tribal Health Services Division’s HCV continuity of care.	Challenges with: - accurately identifying the prevalence of a disease among tribal members. - assessing the prevalence of risk behaviors throughout the population. - reaching individuals who might feel stigmatized and do not desire treatment. Some patients were ashamed of diagnosis and afraid of family or community’s perceptions and reactions. The care of and outreach to HCV patients was complicated by comorbidities and acute situations (e.g., seizures, withdrawals, overdose, domestic violence, etc.).	HCV patients (*n* = 85) were successfully contacted for education, field testing, referral and follow-up, increasing patient outreach in 2009 by 300%. The tribal Health Services Division experience provides an HCV model for other tribes or rural and underserved populations to replicate.

^1^ CLD—Chronic liver disease; ^2^ HCV—Hepatitis C virus; ^3^ ALD—Alcohol-related liver disease; ^4^ NAFLD—Nonalcoholic fatty liver disease; ^5^ HCC—Hepatocellular carcinoma (liver cancer).

## Data Availability

Available upon request from the corresponding author.

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
