# Peer review of "Advancing Liver Cancer Prevention for American Indian Populations in Arizona: An Integrative Review"

_ijerph, 2022, doi:10.3390/ijerph19063268_

Round 1

Reviewer 1 Report

This is a timely article and is well-written. My only two concerns while going through the article is about the organization.

1) Why is there a section on the background and then introduction? It seems like these two sections are connected and should not be separated by two. 

2) In the abstract, methods section: I would have liked to see a few more details about the search strategy such which databases were used and what the inclusion search criteria.

Author Response

Reviewer #1

Comment: Why is there a section on the background and then introduction? It seems like these two sections are connected and should not be separated by two.

Author’s Response: Thank you for this suggestion. We agree, the flow of the article is improved by combining these sections and creates a format like other articles published in IJERPH. Page 2, line 89.

Comment: In the abstract, methods section: I would have liked to see a few more details about the search strategy such which databases were used and what the inclusion search criteria.

Author’s Response: Information about the databases and search criteria have been included and strengthen the abstract. We are appreciative of this recommendation. Page 1, Lines 13-22.

Reviewer 2 Report

Authors described about characteristics of liver tumors in the American Indian (AI) population using a systematic review and meta-analysis in this manuscript.

Major

In the introduction, authors point out the several problems of liver diseases and metabolic diseases and social and insurance back grands, etc. in AI population, these problems had been casing by high morbidity rate of liver cancer and etc. But in results, it was just as we expected, and no new findings could be seen.

I am not make sure, which the authors discuss about AI population in the US or in Arizona.

It my concern, this study was covered the very rare population of AI, and it seems difficult to sympathize with many global readers.

Author Response

Reviewer #2:

Comment: In the introduction, authors point out the several problems of liver diseases and metabolic diseases and social and insurance back grands, etc. in AI population, these problems had been casing by high morbidity rate of liver cancer etc. But in results, it was just as we expected, and no new findings could be seen.

Comment: I am not make sure, which the authors discuss about AI population in the US or in Arizona.

Comment: It my concern, this study was covered the very rare population of AI, and it seems difficult to sympathize with many global readers.

Author’s Response: Thank you for your comments. We believe the revisions we made in response to all reviewer’s comments have addressed your concerns, particularly being more inclusive of global readers.

Reviewer 3 Report

TITLE

In my opinion the title is very interesting.

ABSTRACT

The abstract sounds well.

KEYWORDS

Authors did not correctly report all keywords from MeSH Browser. For example, I checked “Health disparities” on MeSH Browser and I did not find this KW. This is important, in my personal opinion, in order to increase the traceability of this paper (and consequently the possibility of the Journal to be cited by Readers and Stakeholders). I suggest checking all KWs.

SECTION: Introduction

The authors stated that: “There is a 20% survival rate five years after receiving a liver cancer diagnosis, and death rates from liver cancer outpace all other types of cancer”. I suggest dividing the survival rate for the two principal liver tumors, HCC and CCC. If it is not possible due to the paucity of date, please, report this date as an incomplete date with no significant relevance in terms of prognosis.

SECTION: Background

This section is very interesting.

SECTION: Results

Very interesting.

No major concerns.

SECTION: Discussion

This section could be improved to add some paragraph concerning specific problem.

For example: the authors reported the problems of NAFLD and obesity in the American Indian populations. At the state of art, the use of ultrasound in the surveillance program for patients with high risk of developing HCC is recommended. However, it is well known that the ultrasound is limited in cases of obesity and fat livers. To date, the surveillance program and also the future strategy to improve the surveillance program [Non-enhanced magnetic resonance imaging as a surveillance tool for hepatocellular carcinoma: Comparison with ultrasound. J Hepatol. 2020;72(4):718-724. doi:10.1016/j.jhep.2019.12.001   -----    Proposal of a new diagnostic algorithm for hepatocellular carcinoma based on the Japanese guidelines but adapted to the Western world for patients under surveillance for chronic liver disease. J Gastroenterol Hepatol. 2016;31(1):69-80. doi:10.1111/jgh.13150], will allow to overcome the ultrasound limitations by using MRI. Could the Authors discuss these themes?

The authors did not mention treatments strategy. In the era of tailored medicine [J Hepatol. 2019;71(6):1175-1183. doi:10.1016/j.jhep.2019.08.015], is it conceivable that alternative therapeutic strategies can be considered for the populations described?

it is possible to speculate on the fact that some types of treatment such as transplants, or chemoembolization, but also the evaluation of the response with mRECIST require expert operators [Med Phys. 2012;39(5):2491-2498. doi:10.1118/1.3702457 --- Eur Radiol. 2018;28(9):3611-3620. doi:10.1007/s00330-018-5393-3] and therefore very prestigious and probably very expensive reference centers in systems with prevalently private towing. For example: Could the new systemic therapies, relatively easier to use, be reserved for patients even not in advanced states of liver disease?

Author Response

Reviewer #3:

Comment: Authors did not correctly report all keywords from MeSH Browser. For example, I checked “Health disparities” on MeSH Browser and I did not find this KW. This is important, in my personal opinion, in order to increase the traceability of this paper (and consequently the possibility of the Journal to be cited by Readers and Stakeholders). I suggest checking all KWs.

Author’s Response: Thank you for this suggestion. All keywords have been updated to be consistent with the NIH MeSH Browser. Page 1, Lines 27-28.

Comment: The authors stated that: “There is a 20% survival rate five years after receiving a liver cancer diagnosis, and death rates from liver cancer outpace all other types of cancer”. I suggest dividing the survival rate for the two principal liver tumors, HCC and CCC. If it is not possible due to the paucity of date, please, report this date as an incomplete date with no significant relevance in terms of prognosis.

Author’s Response: The survival rates for HCC and CCC have been divided per the reviewer’s suggestion. Page 1, Lines 34-36.

Comment: Discussion. This section could be improved to add some paragraph concerning specific problem. For example, the authors reported the problems of NAFLD and obesity in the American Indian populations. At the state of art, the use of ultrasound in the surveillance program for patients with high risk of developing HCC is recommended. However, it is well known that the ultrasound is limited in cases of obesity and fat livers. To date, the surveillance program and also the future strategy to improve the surveillance program [Non-enhanced magnetic resonance imaging as a surveillance tool for hepatocellular carcinoma: Comparison with ultrasound. J Hepatol. 2020;72(4):718-724. doi:10.1016/j.jhep.2019.12.001   -----    Proposal of a new diagnostic algorithm for hepatocellular carcinoma based on the Japanese guidelines but adapted to the Western world for patients under surveillance for chronic liver disease. J Gastroenterol Hepatol. 2016;31(1):69-80. doi:10.1111/jgh.13150], will allow to overcome the ultrasound limitations by using MRI. Could the Authors discuss these themes?

Authors Response: We provided more information on methods that can be used for surveillance screening. Thank you for this suggestion. Pages 12-13, Lines 369-388.

Comment: The authors did not mention treatments strategy. In the era of tailored medicine [J Hepatol. 2019;71(6):1175-1183. doi:10.1016/j.jhep.2019.08.015], is it conceivable that alternative therapeutic strategies can be considered for the populations described?

Authors Response: Thank you for this suggestion. We included information on the cornerstone treatment strategy for liver disease (e.g., weight loss) within the context of early detection and screening for primary prevention efforts. Lines 385-387.

Comment: It is possible to speculate on the fact that some types of treatment such as transplants, or chemoembolization, but also the evaluation of the response with mRECIST require expert operators [Med Phys. 2012;39(5):2491-2498. doi:10.1118/1.3702457 --- Eur Radiol. 2018;28(9):3611-3620. doi:10.1007/s00330-018-5393-3] and therefore very prestigious and probably very expensive reference centers in systems with prevalently private towing. For example: Could the new systemic therapies, relatively easier to use, be reserved for patients even not in advanced states of liver disease?

Author’s Response: The authors acknowledge there are emerging therapeutic strategies for HCC treatment, particularly with developing pharmacotherapies. However, we do not feel the scope or results of the review inform or support discussing the feasibility, acceptability, or utility of new systemic therapies for primary prevention efforts. We will certainly keep this in mind as we move forward with our work and thank you for this suggestion.